# Gestational Pemphigoid—From Molecular Mechanisms to Clinical Outcomes: A Case Report and Review of Literature

**DOI:** 10.3390/life14111427

**Published:** 2024-11-06

**Authors:** Miruna Ioana Cristescu, Cristina Violeta Tutunaru, Anca Panaitescu, Vlad Mihai Voiculescu

**Affiliations:** 1Department of Dermatology, Carol Davila University of Medicine and Pharmacy, 050474 Bucharest, Romania; miruna-ioana.cristescu@rez.umfcd.ro (M.I.C.); vlad.voiculescu@umfcd.ro (V.M.V.); 2Elias University Emergency Hospital, 011461 Bucharest, Romania; 3Department of Dermatology, University of Medicine and Pharmacy of Craiova, 200349 Craiova, Romania; 4Department of Gynecology, Carol Davila University of Medicine and Pharmacy, 050474 Bucharest, Romania; panaitescu.anca@gmail.com; 5Filantropia Clinical Hospital, 011132 Bucharest, Romania

**Keywords:** gestational pemphigoid, pemphigoid gestationis, histopathology, clinical diagnosis, transplacental passage, immunoglobulins, dermatology, neonatology

## Abstract

Gestational pemphigoid is a rare, autoimmune, subepidermal bullous disease with an incidence of 1 in 50,000 pregnancies, displaying itself through pruritic erythema and urticarial papules and plaques that evolve into tense bullae. Histopathological findings consist of subepidermal vesicles with perivascular eosinophils and lymphocytes, and direct immunofluorescence reveals C3 complement and, more rarely, IgG in a linear band along the basement membrane. The course is usually self-limiting within 6 months after delivery but, later, can be triggered by subsequent pregnancies, menstruation, or treatment with oral contraceptives. The newborn can be affected due to the transplacental passage of the maternal immunoglobulins, but, usually, less than 10% of newborns will develop lesions similar to pemphigoid gestationis. The diagnosis and management pose a difficult challenge and should be guided by the severity of the disease. We, therefore, provide a short literature review and discussion plus a case from our clinic, with a typical presentation but a delayed diagnosis and an undulating evolution, with severe manifestations and particularly difficult management due to unexpected complications.

## 1. Introduction

Gestational pemphigoid is a rare autoimmune skin disorder arising in pregnancy and causing subepidermal blistering [1]. With an incidence of 1 in 50,000 pregnancies, primarily manifesting in the second and third trimesters, it poses unique challenges in diagnosis and management [2]. Multigravidae women are prone to an earlier onset and a longer time to remission [2]. The course is usually self-limiting within 6 months after delivery but can relapse with subsequent pregnancies, menstruation, or oral contraceptives [3].

The pathogenesis resembles the one of bullous pemphigoid, with the presence of IgG autoantibodies against BP180 * and, in part, against BP230 *, constituents of the hemidesmosome, leading to the detachment of the dermis and epidermis and the emergence of blisters, inflammation and erosions [1]. The diagnosis is suggested by the clinical findings such as pruritic erythema and urticarial papules and plaques progressing into papulovesicles and tense bullae, mostly in the periumbilical area, abdomen, trunk, legs, and arms, usually sparing the mucosae [2]. Histopathological findings consist of subepidermal vesicles, with perivascular eosinophils and lymphocytes [4], and direct immunofluorescence reveals C3 complement and, more rarely, IgG deposition in a linear pattern along the basement membrane [1,4]. The therapeutic management is mainly based on the severity of the disease, with first-line treatment consisting of topical or systemic corticosteroids combined with second-generation antihistamines and second-line treatment with alternative or adjuvant therapies such as anti-inflammatory antibiotics, dapsone, and topic immunomodulators [5,6].

The newborns are usually unaffected, but 10% of them will present cutaneous lesions in the form of neonatal pemphigoid, caused by the transplacental passage of maternal antibodies into the fetal bloodstream [1,7,8]. The approach should be multidisciplinary and include a dermatologist, obstetrician, neonatologist, pediatrician, and endocrinologist.

## 2. Clinical Case

We report the case of a 42-year-old woman with 31 weeks of gestation, previously diagnosed with polycystic ovary syndrome, showing intense pruritus and urticarial papules and plaques, tense intact bullae localized on the periumbilical area, trunk, and the extremities, with a duration of three weeks and rapidly progressive onset (Figure 1). Previous to the presentation in our clinic, the patient accessed a few medical care units, including obstetrics-gynecology and internal medicine. The first diagnosis was intrahepatic cholestasis of pregnancy because the rash was absent at the moment, and the patient received treatment with ursodeoxycholic acid. The evolution was unfavorable with the formation of skin sores, and the patient was then referred to the dermato-venereology department with the following potential differential diagnosis: polymorphic eruption of pregnancy, atopic eruption of pregnancy, or gestational pemphigoid.

Upon admission, laboratory tests showed slight increases in leukocytes (11.36 × 10^3^/µL, normal: 4.6–10.2 × 10^3^/µL) and eosinophils (1.77 × 10^3^/µL, normal: 0.0–0.7 × 10^3^/µL; 15.6%, normal: 0.0–7.0%), increased C-reactive protein (14.3 mg/dL, normal: <10 mg/dL) and mixed dyslipidemia. Even though there was a high suspicion of gestational pemphigoid, there was no histopathological or direct immunofluorescence confirmation because the patient did not consent to a skin biopsy. Further recommendations included the ELISA test for circulating IgG antibodies against B180. Considering the symptoms’ severity and the lesions’ extension, systemic corticosteroid treatment was initiated with 30 mg prednisone daily associated with second-generation antihistamine drug cetirizine and proton-pump inhibitor omeprazole. Local treatment included the topical steroid, hydrocortisone acetate, and the antibiotic erythromycin. The purpose was to maintain the dose until obtaining a clinical response (no arousal of new lesions) with subsequent tapering afterward, but the patient was uncooperative and requested to be discharged after three days against medical advice. It was recommended to continue the systemic steroids at home with gradual withdrawal during the following month (5 mg each week). She was lost to follow-up and we do not have the certainty of her compliance with treatment. The patient was directed to the obstetrics-gynecology department where the pregnancy was monitored by ultrasound with no pathological findings; however, the cutaneous eruption worsened.

Two weeks later, she was again admitted to the obstetrics-gynecology department of our hospital and she gave birth to a premature male newborn with minimal rash consisting of annular, erythematous-edematous plaques on the anterior trunk, upper limbs, and facial erythema (Figure 2). The neonate was given a single dose of dexamethasone and was clinically monitored. At the time of the delivery, the mother’s eruption was exacerbated and extensive compared with the last presentation, consisting of tense bullae involving the face, palms, and soles, scales disseminated on the whole body, and post-bullous erosions. She was transferred to the dermatology department for diagnosis confirmation and specific treatment. An 8 mm punch skin biopsy was performed with the patient’s informed consent, which revealed a subepidermal blister containing an eosinophilic infiltrate and a mixed perivascular inflammatory infiltrate, with an abundance of eosinophils in the upper dermis, confirming the clinical suspicion of gestational pemphigoid (Figure 3). We mention that the patient performed the ELISA test during this period, with a positive result of IgG antibodies against BP180 in titer 1:5120 (reference range < 1:10). Of note, the patient worked abroad for years, recently moved back in, and had no health insurance, thus the impossibility of performing direct immunofluorescence. The laboratory findings showed leukocytosis (15.27 × 10^3^/µL, normal: 4.6–10.2 × 10^3^/µL) with neutrophilia (9.81 × 10^3^/µL, normal: 1.5–6.9 × 10^3^/µL) and normal eosinophil count. The patient was claimant, dysphoric, unstable emotionally, and in a state of motor restlessness and mental tension. She also complained about insomnia. Although we explained that the newborn could have a transient rash due to the placental passive transfer of antibodies, the patient insisted that the baby had an infection and was not well fed. A psychiatric consult was requested and it established that she had mixed anxiety–depressive disorder, favored by a preexisting anxious disposition. Subsequently, she received medical treatment with antipsychotics and anxiolytics combined with psychotherapy. Regarding the cutaneous manifestations, the patient underwent 16 mg of dexamethasone iv per day, with progressive dose tapering, and switched to 40 mg of prednisone orally per day, also adding systemic antibiotic treatment with 2 g of ampicillin iv daily. The local treatment consisted of washes with boric acid 1% solution and topical corticosteroids and antibiotics to prevent suprainfection of the post-bullous erosions. The recommendations at discharge included continuing the steroid treatment at home and dose tapering for the next 28 days along with topical corticosteroid and psychiatric treatment. During the follow-up, we noticed a relapse, with new bullae on the neck and face at a dose of 30 mg of prednisone. Given the persistence of the disease after delivery, we considered adding a corticosteroid-sparing agent to circumvent the unpleasant side effects of the long-term corticotherapy. After measuring the glucose-6-phosphate dehydrogenase(G6PDH) level, which showed normal activity, the decision was to administer dapsone 1 mg/kg daily equivalent to approximately 100 mg per day. After one month, the patient was reevaluated and the clinical examination revealed post-inflammatory hyperpigmentation on the previously affected areas, particularly erythematous plaques with numerous millia in the axillary area, cubital fossa, forearms, and hands (Figure 4).

## 3. Discussions

Gestational pemphigoid should be taken into consideration in the clinical setting of cutaneous eruption during pregnancy, especially when associated with itching. The symptom is usually interpreted as a common, harmless complaint in pregnant women [4]. In the absence of the typical rash, it can be easily misdiagnosed. In our case, the onset of the disease was marked solely by itching, which led to multiple presentations and a delayed diagnosis. For our patient, it would have been important to obtain a skin biopsy at the initial presentation to confirm the gestational pemphigoid diagnosis and provide adequate treatment. We decided to obtain confirmation through a serology sample and recommended the patient perform the ELISA test, which was not available at the hospital laboratory. ELISA test determines circulating IgG antibodies against BP180, is sensitive and specific for gestational pemphigoid, and is used as a screening test or to monitor the disease activity [9]. The differential diagnosis alternatives included polymorphic eruption of pregnancy and atopic eruption of pregnancy. We excluded the intrahepatic cholestasis of pregnancy based on the laboratory findings (absence of elevated bilirubin levels, no hepatic cytolysis, and normal prothrombin time). In the first two cases, during the initial phases, the clinical picture is similar and the histologic findings can resemble the ones in gestational pemphigoid, but the direct immunofluorescence is negative [1,10]. Based on the disposition and aspect of the lesions, with a negative history of atopy, we ruled out the possibility of atopic eruption of pregnancy. For the remaining options, the management strategy is similar, with the aim of alleviating the pruritus and preventing the formation of new lesions. When choosing the strategy, the clinician must consider the severity of the cutaneous eruption and the moment of pregnancy, whether the patient is pregnant or postpartum. The timing is important because it influences the corticotherapy dosage or the choice between steroid-sparing drugs. The management in our case was influenced by the patient, leading to a delayed diagnosis, inadequate surveillance, and insufficient medication during gestation.

Regarding the acute psychiatric manifestations, in our opinion, a preexisting anxious disposition favored them. Therefore, in the pregnancy setting, characterized by hormonal changes in the HPA axis and with the addition of corticosteroid medication, an anxious depressive syndrome might be triggered [11]. It is known that HPA axis regulation is influenced by cortisol, ACTH, and CRH, but in women with postpartum depression, studies have identified a hyperactive HPA axis [12]. The triggers are reported to be heritable, having a genetic predisposition, or being influenced by life events [13]. The acute symptoms occurred postpartum, which is characteristic of postpartum depression in women with the highest incidence in the first three months after giving birth [14,15].

The outcome is positive because usually gestational pemphigoid resolves spontaneously after pregnancy, but it can present an exacerbation after birth, as in our case. It is important to inform the patient about the predicted evolution of the disease and the possible complications. After an episode, the lesions take an average of 4 to 16 weeks to resolve, typically without scarring [16]. It should also be mentioned that recurrences may occur with subsequent pregnancies or hormone variations, as seen during menstruation or taking oral contraceptives [2]. A particularity of the case was the progression to multiple millia, which can be mistaken as a reactivation of the PG. Few reports describe the evolution of areas formerly affected by bullous lesions [17]. The clinician should be aware of this possibility and make the distinction between the two entities.

### 3.1. Review

We conducted online research using Medline (PubMed) online repository for reviews written in English, using keywords such as “pemphigoid gestationis”, “gestational pemphigoid”, and “herpes gestationis”. Articles were chosen for the pertinence to our study, omitting those without direct significance. From the initial 752 articles reviewed, we concentrated our analysis on 101 review cases made public in English since the first case was published in 1946 [18].

### 3.2. Definition

Gestational pemphigoid (pemphigoid gestationis, GP) is an uncommon, pruritic, autoimmune blistering skin disorder that occurs gestationally [1]. Herpes gestationis was the first term used to describe PG due to the clinical vesicular feature of the eruption [18,19,20].

GP is characterized by pruritus and an eruption described as erythema multiforme-like or urticarial plaques, with tense blisters emerging in the periumbilical region that can spread to the limbs [21,22]. The onset is during the second or third trimester of the pregnancy, with rare cases manifesting in the first trimester and after birth. Primigravidae have lesser chances to develop PG in comparison with multigravidae. In the latter case, gestational pemphigoid symptoms become visible sooner compared with the former and remission takes longer [1,23]. Furthermore, succeeding pregnancies may lead to more serious skin eruptions [24,25]. There are also some risks regarding gestation such as inadequate fetal development, premature birth, and even neonatal cutaneous eruption and low birth weight [26].

### 3.3. Epidemiology

Gestational pemphigoid occurs rarely but is a specific condition tied to gestation. The incidence is reported to be approximately 1 in 50,000 pregnancies [21]. There is no race difference or geographic distribution reported. The onset age ranges between 17 and 41 years. PG can develop during any pregnancy, but most likely (63 to 75% of cases) during the first three [27]. Having a history of PG in a previous pregnancy represents a critical recurrency trigger, with patients presenting an earlier onset and increased symptoms severity [28].

### 3.4. Etiopathogenesis

The etiology of gestational pemphigus remains largely unknown. Despite being usually associated with the first gestation, PG can accompany some disorders resembling pregnancy such as hydatidiform mole, choriocarcinomas, and trophoblastic tumors, as mentioned in some cases [25,29,30].

The pathogenesis resembles the bullous pemphigoid one since PG patients are also positive for auto-antibodies against BP180 and partially against BP230. These two hemidesmosomal proteins are parts of the dermo–epidermal junction [20,30,31]. There is evidence that four out of five distinct epitopes expressed on BP180 are principal antigenic sites [28,30]. The autoimmune conflict starts in the placenta in the first trimester where placental tissues such as trophoblastic cells and amnio chorionic stromal cells express BP180. These structures are not recognized by the maternal immune system as self, thus triggering an immune response [21,29]. IgG antibodies against these structures are formed, which have a cross-reaction with the BP180 proteins that are found in the skin [25,30,31]. This results in complement activation through the classical pathway and subsequent chemoattraction of eosinophils with degranulation [32,33,34]. The proteolytic enzymes that are released from eosinophilic granules dissolve the bond between the epidermis and dermis with subsequent blister formation [24,32]. Some authors suggest that some paternal antigens located on the chorionic villi belonging to the second class of the major histocompatibility complex (MHC) may start an immune maternal response leading to antibodies directed against the amniotic basement membrane. Furthermore, these antibodies could cross-react with different maternal skin antigens causing maternal and even fetal disease. Some studies sustain a genetic predisposition since there is a correlation related to PG and the second-class human leukocyte antigen (HLA), particularly the phenotype HLA-DR3/HLA-DR4; still, there is not a well-known catalyst in resulting these autoantibodies [32,35]. The maternal HLA-DRB1*0301 (HLA-DR3) and HLA-DRB1*0401/040X (HLA-DR4), belonging to the major histocompatibility complex (MHC) class II reveals the critical role of autoimmunity in the pathogenesis of the PG due to their predominant correlation with the disease [1,29]. The MHC class II antigens are not usually expressed on the trophoblast; thus, the fetus is protected from the maternal immune system under normal conditions [33]. Human leukocyte antigens PG3 and PG4 are also frequently correlated with PG [1,29].

Maternal antibodies can passively transfer through the placenta and can result in neonatal pemphigoid (NP) in approximately 10% of affected gestations. PG seems to be the result of immunoglobulin G (IgG) autoantibodies directed against the NC16A domain of collagen XVII in most cases (90%) [34,36]. A possible explanation for the improvement of the condition just before birth is an increased progesterone level during the last few weeks of gestation [33].

### 3.5. Clinical Course and Features

Even though it has some common clinical manifestations with the majority of autoimmune bullous disorders, PG often has an irregular presentation [24,37]. Obtaining a thorough history is crucial since this rare condition exhibits an unpredictable clinical course, characterized by episodes of exacerbation and remission, with flare-ups being common during the peripartum and postpartum periods [29,38].

Gestational pemphigoid usually occurs in the second or third trimester of gestation, with severe itching being the major symptom [28,39]. The typical eruption composed of erythematous urticarial papules and plaques initially appears on the abdomen, starting in the periumbilical area, and later extending to other parts of the trunk and the extremities [40]. Usually, the face and mucosal areas are unaffected. In a matter of days and up to 4 weeks, the primary cutaneous eruption evolves into vesicles or tense bullae on an erythematous base [28]. Gestational pemphigoid typically resolves within six months postpartum, but in some scenarios, such as antibody increased level or female sex hormone variation, the condition may linger or aggravate. Also, PG can reoccur due to the following: menstruation, subsequent gestations, or the use of birth control pills based on estrogens and progesterone. GP has been linked to a higher probability of Graves’ disease, in the context of common genetic factors and immune system variations during gestation, but there are insufficient data supporting their association [31].

Adverse pregnancy outcomes (APO) associated with GP include premature delivery and impaired fetal development with decreased gestational age, reported in 34 to 37% of cases [16,41]. There is no evidence suggesting an increased risk of spontaneous abortion. Cutaneous bullous eruption and the early appearance of the PG are linked to the obstetrical outcome according to the literature findings [32,41]. An unfavorable outcome is connected to the development of the PG early in the pregnancy during the first and second trimesters [23,32]. A study shows that fetal growth restriction is influenced by anti-BP180 IgG antibody ELISA values, with a cutoff higher than 150 IU [41]. The risk assessment for APO includes the clinical markers and the anti-BP180 antibody ELISA values as predictors [41].

### 3.6. Diagnosis

The diagnosis of PG is based on the clinical aspects, histological features, direct immunofluorescence (DIF) or indirect immunofluorescence (IIF), enzyme-linked immunosorbent assay (ELISA), and C4d immunochemistry technique [23,30,42].

The gold standard investigation for confirming gestational pemphigoid is DIF staining on skin biopsy specimens [43]. The characteristic findings on DIF include linear deposits of C3 and IgG autoantibodies in the basement membrane zone [26,44]. Since the presence of C3 deposits is a constant finding throughout all DIF stains, it is regarded as pathognomonic for PG. Notably, DIF results can remain positive from up to 6 months to 4 years after clinical remission [29].

Histopathologically, initial lesions present papillary edema with lymphocyte infiltration and eosinophilic infiltrate in the dermis [3].

The classical histopathological features of the bullous stage include subepidermal blister formation, accompanied by perivascular lymphocytic and eosinophilic infiltrates, as well as mild basal spongiosis [44,45]. However, these histological characteristics are nonspecific and can also be observed in other dermatoses, such as polymorphic eruption of pregnancy [44]. However, invasive procedures in childbearing women should be minimized, used cautiously, and complemented with noninvasive diagnostic methods such as indirect IIF and ELISA. In patients with gestational pemphigoid, IIF detects circulating IgG antibodies in 30–100% of cases. In comparison to DIF, IIF has lower sensitivity and specificity and, thus, is not a reliable diagnostic tool [23,46]. The BP180-NC16a ELISA assay detects circulating IgG antibodies targeting the 16th non-collagenous domain (NC16a) of BP180 [10]. The presence of NC16a antibodies is determined by indirect ELISA, which utilizes horseradish peroxidase-conjugated antihuman IgG. Tetra-methyl-benzidine represents the substrate for horseradish peroxidase, determining a change in color proportional to the quantity of anti-NC16a antibodies. This test is considered to have high sensitivity and specificity. An advantage is the noninvasive nature (no skin specimen needed) and the time efficiency (results within 3.5 h). In contrast to DIF and IIF, serum autoantibody levels measured via ELISA have been found to correlate with disease severity [43].

### 3.7. Differential Diagnosis

The main differential diagnosis for PG includes all the pregnancy-specific dermatoses: polymorphic eruption of pregnancy (PEP), atopic eruption of pregnancy (AEP), intrahepatic cholestasis of pregnancy (ICP) and pustular psoriasis of pregnancy (PPP) [30,47,48,49]. PEP (polymorphic eruption of pregnancy) usually develops in multiple gestation pregnancies and is possibly associated with increased maternal weight gain [43]. There are many similarities between PEP and PG, such as outbreaks of symptoms, affected areas, intense itchiness, and urticarial aspects. Distinguishing between these two skin disorders is challenging in the initial stage (pre-bullous) of PG, regarding the clinical manifestations and histology, respectively. A negative DIF result suggests the polymorphic eruption of pregnancy diagnosis [1,10,48]. AEP, compared with other pregnancy dermatoses, occurs early in the pregnancy—more frequently during the first or second trimester. Its diagnosis is based mainly on the clinical features due to the lack of laboratory investigations [47]. No specific features are revealed by skin biopsy and DIF and IIF are negative, while total IgE antibody serum levels might be increased. Remission of AEP lesions appears early in the after-birth period [1,28,37]. Intrahepatic cholestasis of pregnancy presents with pruritus as the main symptom [1]. The difference between intrahepatic cholestasis of pregnancy and PG is based on laboratory investigations: bile acids (>10 µmol), prothrombin time, liver enzymes, and most notably serum anti-BP180 antibodies. There is a high specificity and sensitivity (96 to 100%) of anti-BP180 antibodies for the diagnosis of PG. The distinction between these two disorders is vital since the main treatment for ICP consists of ursodeoxycholic acid while in PG, topical corticosteroids are used [32]. Pustular psoriasis of pregnancy (PPP)typically presents in the third trimester of gestation and can recur in subsequent pregnancies. Clinically, numerous sterile pustules in a circinate disposition appear in the intertriginous areas and extend centrifugally. Histology shows spongiform pustules with neutrophils, psoriasiform hyperplasia, and parakeratosis, while DIF is negative [47]. Differential diagnosis of PG can also be performed with urticaria, allergic contact dermatitis, impetigo, and bullous drug eruption [29,37,45].

### 3.8. Treatment

Treatment of PGvaries according to the stage of the cutaneous eruption and its extent [29]. The therapy aims to reduce itching, slow the progression, and prevent the emergence of new bullae [1,5,19]. Based on case studies, the first-line treatment consists of topical or systemic corticosteroids along with second-generation antihistamines. In mild cases with minimal blistering, topical steroids and antihistamines can be sufficient [5,19,29]. Mild- or moderate-potency topical corticoids are the treatment of choice in pregnancy [1,5]. All topical corticosteroids, except very potent ones, can be used in postpartum and during breastfeeding without being directly applied to the nipple area as increased ingestion can cause a newborn’s hypertension [46,50]. Calcineurin inhibitors up to 5 g/day for a couple of weeks represent the treatment option when steroids are contraindicated [40]. Systemic corticosteroids are recommended when more than 10% of the body is affected or the topical treatment is insufficient. Both prednisolone and prednisone are inactivated by the enzyme 11-β-hydroxysteroid dehydrogenase found in the placenta, leading to a lower concentration of the medication reaching the fetus [1,29]. Corticosteroid use during pregnancy is considered safe, with no teratogenic effect, but might present a risk for growth restriction and prematurity at higher doses [51]. In a study on pregnant women with rheumatoid arthritis, a more frequent disease that requires corticotherapy, the results showed no significant correlation between glucocorticoid administration (dose < 10 mg of prednisone or equivalent) and low birth weight [52]. At the moment, it is debatable whether the APO seen in patients treated with glucocorticoids is determined by the medication or the underlying condition [51]. The principles for prescribing corticotherapy in this case should be based on using the minimal effective dose and the shortest duration to control disease manifestations. For GP, prednisolone 0.5 mg/kg daily but less than 20 mg per day is the preferred regimen during pregnancy [29,53]. In severe PG, up to 2 mg/kg per day of prednisone can be administered in the after-birth period [32].

The second-line treatment with alternative or adjuvant therapies consists of anti-inflammatory antibiotics, dapsone, topic immunomodulators, or immunosuppressant agents like cyclosporine or azathioprine [21,22,42,50,53]. Azathioprine is an immunosuppressive drug, which inhibits purine synthesis and is relatively safe in pregnant women if taken at the maximum admitted daily dose of 2 mg/kg. In contrast, administering higher doses opens up the possibility of causing lymphopenia in a child [26,46]. Dapsone or diamino-diphenyl sulfone is an antibiotic with anti-inflammatory capabilities. Breastfeeding is not appropriate during dapsone therapy if the child has G6PD deficiency; otherwise, it is considered safe. G6PD deficiency can cause hemolytic anemia in mother and child [46]. Cyclosporine, a calcineurin inhibitor with immunomodulatory properties, has a C-class indication during pregnancy and should be considered when the benefits exceed the risks in refractory patients. The administration has not been associated with congenital malformations, but premature delivery and low birth weight have been reported. It is important to mention that the drug should be avoided during breastfeeding because it is excreted in milk [54]. Intravenous immunoglobulins (IVIG) have recently been used effectively, either as monotherapy or alongside other treatments, and offer a safer option for both the patient and fetus compared with immunosuppressants [33,55]. Rituximab represents another treatment choice in preventing recurrent PG, but it is recommended to wait about one year before conception due to the medication’s very long half-life [6,40]. The combination of rituximab and IVIG is a good option in nonresponsive patients [6,33]. Still, in the postpartum/post-abortion period, immunosuppressant/immunomodulating agents should come first before rituximab and IVIG due to higher costs [19]. Plasmapheresis is a procedure by which pathogenic antibodies are removed from a patient’s plasma. It is considered safe in pregnancy and can be an alternative method of choice in case of severe or refractory PG [46]. Theoretically, a new drug for PG is Dupilumab, a biologic therapy successfully used for atopic dermatitis and prurigo nodularis. It inhibits the alpha subunit of the interleukin 4 receptor and improves the pruritus and the bullous eruption, characteristic of PG [56,57]. Recently, it has been used in cases of refractory and corticosteroid-dependent bullous pemphigoid. Even more, it has not been linked with pregnancy complications such as major birth defects, miscarriage, or poor maternal and fetal outcomes [57].

### 3.9. Outcomes

Usually, gestational pemphigoid resolves spontaneously after pregnancy, but it can present an exacerbation after birth. After this episode, the lesions take an average of 4 to 16 weeks to resolve, typically without scaring [16,48]. Recurrences may occur with subsequent pregnancies or hormone variations, as seen during menstruation or taking oral contraceptives [2]. In the literature, we can find data about some particular cases that evolve with the formation of multiple millia in areas of previous blistering [17]. There seems to be an immunological predisposition, explained by the human leukocyte antigen DQ6, which can be related to millia formation in patients with bullous pemphigoid [58]. When facing this particular clinical picture, it is important to differentiate between a possible relapse of the disease and the presence of multiple millia secondary to the regeneration process.

The neonate might be affected due to the transplacental passage of immunoglobulins. About 10% of infants are reported to present mild skin lesions, which typically resolve within a few days [16]. The structure of the IgG molecule and the specific receptor play an important role and will be further discussed for a better understanding of the process.

Immunoglobulin G (IgG) is the most prevalent in human serum of the five isotypes. It presents four subclasses, IgG1, IgG2, IgG3, and igG4, with different structures consisting of variations in their constant region, their hinges, and upper CH2 domains. This further translates into different functions of these immunoglobulins, such as phagocytosis, antibody-dependent cell-mediated cytotoxicity, or complement activation. They are composed mostly of protein (82–96%) and carbohydrates (4–18%), mostly similar at this level. The molecule structure is composed of two heavy chains (50 KDa) and two light chains (25 kDa), bonded together by disulfide bonds, in a Y-like shape. The Fc part of the antibody that represents the binding site for the neonatal Fc receptor (FcRn) is formed by the lower hinge region and the CH2/CH3 domains. This is responsible for the placental passage, transport of IgG, and prolonged half-life of IgG [59].

The FcRn receptor is a heterodimer related to the MHC I class and is associated with β2microglobulin. It is made of an α heavy chain that is linked non-covalently with the β2microglobulin light chain [60]. The main ligands for FcRn are IgG and albumin. It interacts with the Fc portion of the IgG and helps with the recycling and transcytosis of IgG within cells, which can also transport IgG across cellular barriers. This results in the characteristic effects of the interaction mentioned before. FcRn is expressed in multiple tissues, in different organs of the body, such as skin microvasculature, and retinal and placental endothelial cells [60,61].

The passage from maternal to fetal blood in humans must pass a histological barrier composed of two cell layers: the syncytiotrophoblasts and endothelial cells of the fetal capillaries. This barrier has the role of filtering substances that pass across it to reach the fetal blood. Nutrients and solutes are transported passively or actively to ensure the physiological growth and development of the fetus. Low molecular weight compounds such as amino acids and ions simply diffuse through the placental tissue. Immunoglobulins, such as IgG, with a higher molecular weight pass using specific transport [62]. The only antibody class that can cross the histological layers of the placenta is IgG, which weighs approximately 160 kDa [62].

The interaction between IgG and the FcRn is influenced by the pH of the environment. At physiologic pH, the affinity is low, whereas at acidic pH < 6.5, the affinity is higher. The transport process is represented by transcytosis, in which endosomes are formed and IgG molecules are internalized. The FcRn receptor is present on the internal surface of the endosomes where the acidic pH favors the binding of IgG to FcRn in complexes. The endosome moves toward the basolateral membrane, to the fetal side, where the near-natural pH enables IgG dissociation from FcRn and releases IgG into the fetal circulation [61,63].

During normal pregnancy, immunoglobulins are naturally transferred from the mother to the fetus to offer crucial protection during the first initial months of life when the newborn’s humoral response is underdeveloped [63]. Conversely, harmful antibodies may cross the placenta and cause transitory autoimmune diseases in the neonate [62].

An active transplacental passage of immunoglobulins kicks off early, at 13 weeks of gestational age, followed by an increase in fetal IgG levels with gestational age, which rise over the plasmatic maternal IgG levels at birth [63]. The antibody transfer through the placenta is also influenced by the immune status of the mother, with specific antibody levels acquired postimmunization, placental integrity, or associated infections. Considering these variables, there are different surveillance protocols for specific situations such as maternal autoimmune or alloimmune conditions where pathologic antibodies are passed into the fetal circulation resulting in complications [63].

Serum levels of maternal immunoglobulins are directly linked to the placental transfer and neonatal IgG levels. In the case of maternal hypergammaglobulinemia, when maternal IgG levels are too high, the FcRn receptors become oversaturated and, subsequently, unbound IgG molecules, leaving them to be destroyed. The result is a decreased fetal antibody transfer [63].

Neonatal autoimmune blistering disease where neonates are affected due to the transplacental transport of IgG autoantibodies occurs with low incidence. The evolution is normally transient, with total resolution after a few weeks, depending on the maternal antibodies’ washout from the neonate’s blood. In the case of gestational pemphigoid, the maternal BP180 antibodies can pass the placental passage and induce neonatal PG in 10% of cases. The effects on the placenta are less studied, but it has been shown that mild villitis and collections of immature villi might be present. The direct effect of the IgG antibodies is on the chorionic cells, resulting in the detachment of basal membrane and undeveloped hemidesmosomes, which can lead to placental malfunction [64]. The results could be minor placental insufficiency, fetal growth restriction, preeclampsia, or premature delivery [1,4,7]. The base mechanism is the autoimmune reaction against placental collagen VII, which is part of hemidesmosomes. A study suggests that a noninflammatory mechanism consisting of decreased expression of collagen XVII in keratinocytes might contribute to the effects [64].

## 4. Conclusions

Even though it is a rare disease, gestational pemphigoid should be suspected in pregnant women presenting blistering and itching. It is important to mention that the diagnosis should be made by a dermatology specialist and the patient needs to be closely monitored.

Some cases, as illustrated here, require inpatient care in order to control the course of the disease and prevent further complications. A multidisciplinary team composed of a dermatologist, pathologist, gynecologist, and neonatologist is essential to properly manage such patients. In our particular case, the role of a psychiatrist was crucial, given that the patient was unwilling to collaborate throughout the hospital stay. Regarding the treatment, we based our decision of administering systemic corticosteroids from the beginning considering the severity and extension of the lesions and later switching to a steroid-sparing treatment to address the refractory skin lesions. Our patient’s particularity was represented by the rapid onset of symptoms and a low-responding disease with a distinct recovery with the formation of multiple millia.

In all cases, the newborn needs to be carefully monitored and both the mother and the neonate should be evaluated regarding possible side effects associated with the administration of corticosteroids. The mother should also be reassured about the transient nature of the condition in the newborns. Furthermore, mothers with a history of gestational pemphigoid should be informed about the elevated risk of relapse in subsequent pregnancies and the possibility of relapses associated with ovulation, menstruation, or the use of oral contraceptives. Although it is more of a cosmetic concern, the possibility of developing multiple millia at any disease phase should be acknowledged.

In conclusion, gestational pemphigoid requires clinical vigilance and a complex approach, with consideration of psychiatric factors, to optimize patient outcomes. This article contributes to understanding the varied clinical courses and challenges that can arise when managing this condition.

## Figures and Tables

**Figure 1 life-14-01427-f001:**
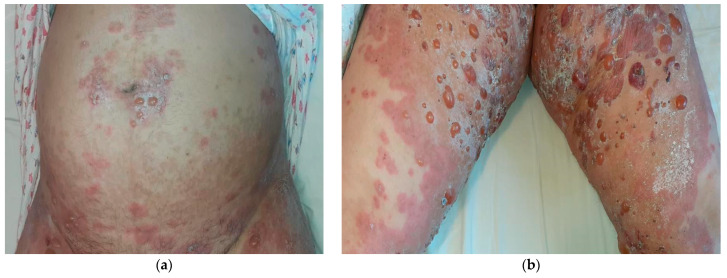
Clinical aspects postpartum:(**a**) urticarial plaques and tense bullae, with classic involvement of the umbilical area; (**b**) tense bullae on erythematous plaques affecting the thighs.

**Figure 2 life-14-01427-f002:**
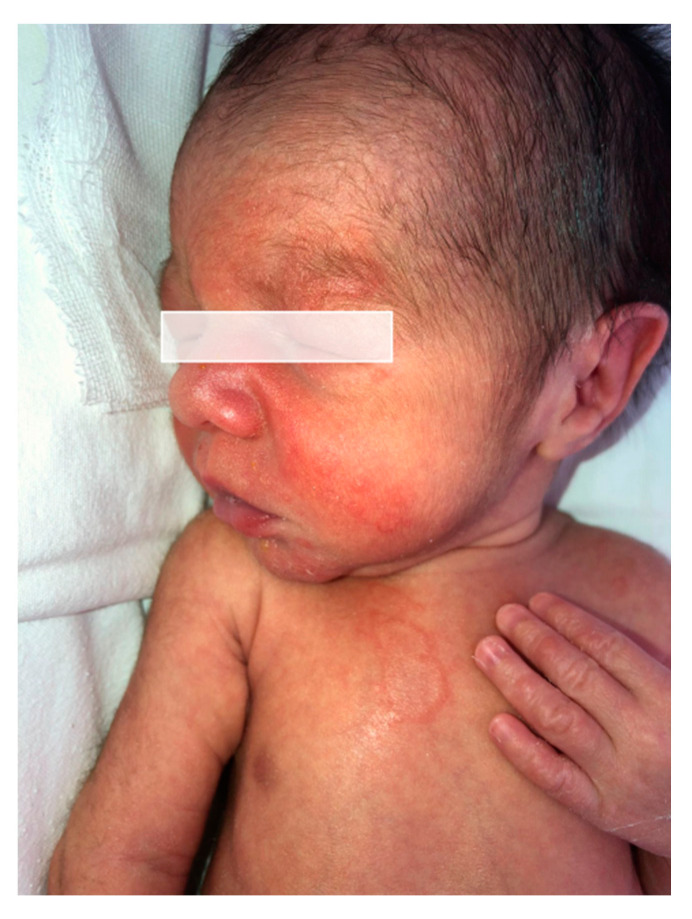
**Newborn lesions:** annular erythematous-edematous lesions on the face and the trunk.

**Figure 3 life-14-01427-f003:**
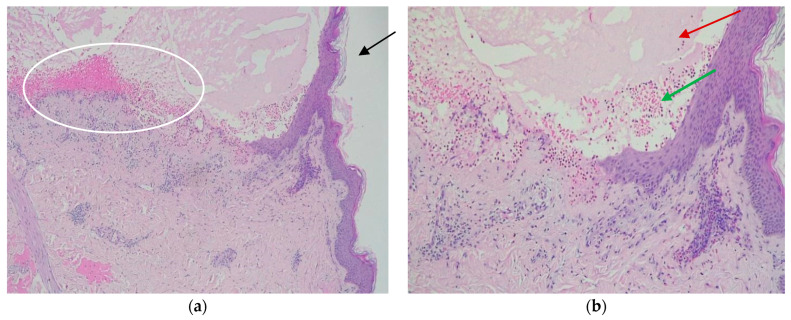
**Histopathological examination:** (**a**) hematoxylin–eosin staining 5×: subepidermal tense dome-shaped bulla, containing serum and eosinophilic infiltrate; upper dermis with edema and perivascular inflammatory infiltrate composed of lymphocytes and eosinophils (white circle); moderate orthokeratosis (black arrow); (**b**) hematoxylin–eosin staining 10× (detail): bullae containing serous exudates (red arrow) and eosinophilic infiltrate (green arrow).

**Figure 4 life-14-01427-f004:**
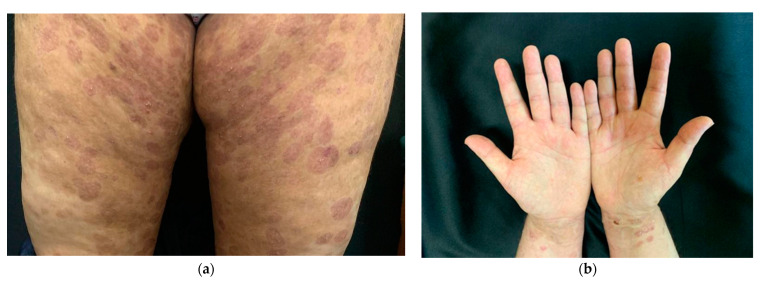
**Clinical aspects at reevaluation after one month:** (**a**) post-inflammatory hyperpigmentation with millia on the thigh region, one month after discharge; (**b**) resolution of the palmar lesions, with some erythematous plaques with multiple milia on the surface located in the wrist region.

## Data Availability

No new data were created or analyzed in this study. Data sharing is not applicable to this article.

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
