# Peer review of "Gestational Pemphigoid—From Molecular Mechanisms to Clinical Outcomes: A Case Report and Review of Literature"

_life, 2024, doi:10.3390/life14111427_

Round 1
Reviewer 1 Report
Comments and Suggestions for Authors
This is an interesting case report on gestational pemphigoid complicated by low compliance.
1. The first page of the Discussion contains too many repetitions of the case history and perhaps do not belong here. Preferred tests, diagnostic criteria belong in the introduction and details around which laboratory performed tests belongs in the case history, while the Discussion should focus on similar cases, what physicians should be aware of, etc. The details around IgG from the second part of the Discussion also seem better placed in the Introduction, since this is text book knowledge (and is perhaps too detailed), not a discussion. Actually the Discussion has too little clinical discussions.
2. The Conclusion is too long and has little reference to what is mentioned in the Discussion. The conclusion should represent a summary of the main points from the Discussion, not a completely separate text.
3. At the beginning of the discussion you mention that itching is often considered just a symptom of pregnancy - this gives a wrong impression that only itching is classical for gestational pemphigoid. It is more typical that there are skin lesions, not just itch, it is your patient that presented with itch initially.
4. You might mention that while cyclosporine can be given during pregnancy it is contraindicated if breastfeeding. Also, mention if prednisolone is safe during breastfeeding (since you mention its safety during pregnancy).
5. There are multiple typos, please see attachment where these are marked.

Author Response
Thank you for your suggestions! We have reconsidered the structure of our paper and introduced a review part in the discussion section. We kept the conclusions regarding our case at the end.
4. You might mention that while cyclosporine can be given during pregnancy it is contraindicated if breastfeeding. Also, mention if prednisolone is safe during breastfeeding (since you mention its safety during pregnancy).
Response: We have taken into consideration you suggestion and therefore we added information regarding cyclosporine administration during pregnancy and breastfeeding period (lines 386 to 391).
5. There are multiple typos, please see attachment where these are marked.
Response: Thank you for pointing this out. We revised our manuscript accordingly and clarified the marked sections (line 74, lines 82-88, figure 2).
Reviewer 2 Report
Comments and Suggestions for Authors
It is my great pleasure to have an opportunity to review this interesting article. The authors presented a case of intractable gestational pemphigoid and reviewed the disease. I have some comments and questions below.
1. Both "gestational pemphigoid" and "pemphigus gestationis" are used in the text. Either should be used exclusively.
2. As the authors stated in the text, gestational pemphigus is a very rare disease. However, I couldn't find any novel findings in this paper. What the authors consider critical in this case should be emphasized more.
3. The dose of dapsone administered should be described.
4. The basic explanation of immunoglobulins in the discussion is too long. It should be shortened, and the diagnosis, treatment, and pathomechanisms of gestational pemphigoid should be discussed in more detail.
Comments on the Quality of English Language1. There are some grammatical errors. English proofreading should be done.
2. There are many typos listed below. They should be corrected.
pemfigoid > pemphigoid (lane 23), post-bulous> post-bullous (lane 94), phagochytosis > phagocytosis (lane 212), cand > can (lane 225), transplacentar transplacental transfer > transplacental transfer (250), transferend > transferred (lane 256)
Author Response
We would like to thank you for your kind response and suggestions. We have decided to redesign our article and introduced a review part in the discussion section, keeping the conclusions regarding our case.
1. Both "gestational pemphigoid" and "pemphigus gestationis" are used in the text. Either should be used exclusively.
Response: We have modified accordingly and kept the term "gestational pemphigoid" , as presented in the title.
2. As the authors stated in the text, gestational pemphigus is a very rare disease. However, I couldn't find any novel findings in this paper. What the authors consider critical in this case should be emphasized more.
Response: We agree with your point of view. We based our paper on the difficult management of the case, considering the non-compliant patient (refuse to obtain a skin biopsy, non-adherence to treatment, early discharge), the subsequent complications (extended eruption during the third trimester, neonatal cutaneous eruption) and associated conditions (anxiety-depressive disorder). We intended to present the evolution of the patient and emphasize the challenges that may arouse.
3. The dose of dapsone administered should be described.
Response: We have considered your suggestion and we added the information pertaing to dapsone dose in our case. (line 126)
4. The basic explanation of immunoglobulins in the discussion is too long. It should be shortened, and the diagnosis, treatment, and pathomechanisms of gestational pemphigoid should be discussed in more detail.
Response: We included this part into the review section where we consider it fits better.
Finally, we went through the text and emended the grammar errors and typos.
Reviewer 3 Report
Comments and Suggestions for Authors
The aim of this paper, entitled “ Gestational pemphigoid: From molecular mechanisms to clinical outcome” is to present a case with a typical presentation, discussing the pathogenetic, diagnostic and therapeutic issues of this rare disease.
The case report is interesting and well-documented; however, there are some critical issues that should be addressed before the publication, as follows:
The title is misleading: it should be better specified as a clinical case.
It is not clear to which time interval the clinical images refer, this should be specified.
Is the appearance of psychiatric symptoms related to steroid therapy or is it an unrelated or pre-existing event? This aspect should be better clarified and discussed, also with the support of the literature.
The introduction and discussion are excessively long. They should be reduced, avoiding repeated parts.
There are a few typos, that should be corrected.
Comments on the Quality of English LanguageThe aim of this paper, entitled “ Gestational pemphigoid: From molecular mechanisms to clinical outcome” is to present a case with a typical presentation, discussing the pathogenetic, diagnostic and therapeutic issues of this rare disease.
The case report is interesting and well-documented; however, there are some critical issues that should be addressed before the publication, as follows:
The title is misleading: it should be better specified as a clinical case.
It is not clear to which time interval the clinical images refer, this should be specified.
Is the appearance of psychiatric symptoms related to steroid therapy or is it an unrelated or pre-existing event? This aspect should be better clarified and discussed, also with the support of the literature.
The introduction and discussion are excessively long. They should be reduced, avoiding repeated parts.
There are a few typos, that should be corrected.
Author Response
We want to thank you for your suggestions regarding our paper.
We have revised it and reconsidered the category, therefore we added a review part in the discussion section. In this case, the title is more representative for the content.
We have taken into consideration your suggestion and changed the captions of the figures accordingly (Figure 1,2,3,4).
We also emended the text in terms of grammar and typos.
In our case, it was established together with the psychiatric specialist that the onset of the psychiatric manifestations was favored by a pre-existing anxious disposition. Therefore, in the pregnancy setting, characterized by hormonal changes in the HPA axis and with the addition of corticosteroid medication, an anxious-depressive syndrome might be triggered (1). It is known that HPA axis regulation is influenced by cortisol, ACTH and CRH, but in women with postpartum depression studies have shown a hyporeactibe HPA axis (2). The triggers are reported to be heritable, having a genetic predisposition or being influenced by life events (3).The acute symptoms occurred postpartum, which is characteristic of the entity of postpartum depression in women with the highest incidence in the first three months after giving birth (4,5).
1. Alturaymi MA, Almadhi OF, Alageel YS, Bin Dayel M, Alsubayyil MS, Alkhateeb BF. The Association Between Prolonged Use of Oral Corticosteroids and Mental Disorders: Do Steroids Have a Role in Developing Mental Disorders? Cureus. 2023 Apr 15;15(4):e37627. doi: 10.7759/cureus.37627. PMID: 37200642; PMCID: PMC10185922.)
2. Jolley SN, Elmore S, Barnard KE, Carr DB. Dysregulation of the hypothalamic-pituitary-adrenal axis in postpartum depression. Biol Res Nurs. 2007;8(3):210-222. doi:10.1177/1099800406294598
3. Meltzer-Brody S. New insights into perinatal depression: pathogenesis and treatment during pregnancy and postpartum. Dialogues Clin Neurosci. 2011;13(1):89-100. doi:10.31887/DCNS.2011.13.1/smbrody
4.Carlson K, Mughal S, Azhar Y, et al. Postpartum Depression. [Updated 2024 Aug 12]. In: StatPearls [Internet]. Treasure Island (FL): StatPearls Publishing; 2024 Jan-. Available from: https://www.ncbi.nlm.nih.gov/books/NBK519070/
5. Alshikh Ahmad H, Alkhatib A, Luo J. Prevalence and risk factors of postpartum depression in the Middle East: a systematic review and meta-analysis. BMC Pregnancy Childbirth. 2021;21(1):542. Published 2021 Aug 6. doi:10.1186/s12884-021-04016-9)
Round 2
Reviewer 2 Report
Comments and Suggestions for Authors
The authors responded to all comments and questions properly. I don’t have any more comments.
Reviewer 3 Report
Comments and Suggestions for Authors
All my previous comments has been addressed.
The paper can be published in this form
Comments on the Quality of English LanguageAll my previous comments has been addressed.
The paper can be published in this form